# Comparison of Thyroid Hormone Levels between Women Farmers and Non-Farmers in Banten Indonesia

**DOI:** 10.3390/ijerph18126618

**Published:** 2021-06-19

**Authors:** Dian Mardhiyah, Wan Nedra Komaruddin, Fasli Nedra Jalal, Sri Wuryanti, Syukrini Bahri, Qomariah Qomariah, Insan Sosiawan, Himmi Marsiati, Legiono Legiono, Harliansyah Hanif, Susan Woskie, Pornpimol Kongtip

**Affiliations:** 1Department of Public Health, YARSI University, Jakarta 10510, Indonesia; 2Department of Pediatric, YARSI University, Jakarta 10510, Indonesia; wan.nedra@yarsi.ac.id; 3Department of Nutrition, YARSI University, Jakarta 10510, Indonesia; fasli.jalal@yarsi.ac.id (F.N.J.); sri.wuryanti@yarsi.ac.id (S.W.); 4Department of Clinical Pathology, YARSI University, Jakarta 10510, Indonesia; syukrini.bahri@yarsi.ac.id; 5Department of Physiology, YARSI University, Jakarta 10510, Indonesia; qomariyah@yarsi.ac.id (Q.Q.); insan.sosiawan@yarsi.ac.id (I.S.); himmi.marsiati@yarsi.ac.id (H.M.); legiono@yarsi.ac.id (L.L.); 6Department of Biomedical Science, YARSI University, Jakarta 10510, Indonesia; harliansyah.hanif@yarsi.ac.id; 7Department of Public Health, University of Massachusetts Lowell, Lowell, MA 01854, USA; Susan_Woskie@uml.edu; 8Department of Occupational Health and Safety, Faculty of Public Health, Mahidol University, Bangkok 10400, Thailand; pornpimol.kon@mahidol.ac.th

**Keywords:** thyroid hormone, pesticides, occupational health, agricultural health, female farmers, endocrine disrupters, agricultural worker

## Abstract

Pesticides are suspected of being endocrine disruptors. This cross-sectional study measured serum samples for levels of thyroid hormones including thyroid stimulating hormone (TSH), triiodothyronine (T3), thyroxine (T4), free T3 (FT3), and free T4 (FT4) among Indonesian female farmers (*n* = 127) and non-farmers (*n* = 127). A questionnaire was used to collect information on demographics and risk factors including work characteristics and frequency, and the use of home and agricultural pesticides. Results showed that there were no significant differences in the distribution of the clinical categories of thyroid levels between farmers and non-farmers except for FT3 and T4. However, in multivariable regression controlling for confounders, FT3 and T4 were significantly higher for farmers compared to non-farmers. In addition, 32% of farmers had clinically low iodine levels and 49% of non-farmers had clinically high iodine levels. We conclude that pesticide exposure may not be as important as iodine intake in explaining these findings. We recommend counseling by health workers about the importance of using iodized salt for farmers and counseling about high iodine foods that need to be avoided for non-farmers.

## 1. Introduction

In 2020, the Minister of Agriculture in Indonesia stated that the production targets of the country had increased to 59.15 million tons for rice, 30.35 million tons for corn, and 1.29 million tons for soybeans [1]. Along with fertilizer, the use of pesticides forms part of farmers’ efforts to control crop pests and increase farm productivity to meet the increasing demands of Indonesia’s growing population as well as its export needs. Data on the use of pesticides in Southeast Asia, including in Indonesia, are still very limited, although from 2015–2020, the number of pesticide trademarks increased by 4380 brands [2].

Little research has been done to identify the risks faced by Indonesian farmers using pesticides. In Semarang Indonesia, 13 of the 50 Indonesian chili farmers studied (26%) reported severe pesticide poisoning and 37 (74%) reported light poisoning by pesticides [3]. In a study of 70 horticulture farmers in Magelang, Central Java, 14.3% reported pesticide poisoning and 34% had a balance disorder as tested by the Romberg Test [4]. According to the Directorate General of Food Crops, in 2013, 19 farmers (47.5%) had poisoning due to pesticides and 17 farmers (42.5%) suffered from anemia in Gombong Village, Indonesia [1].

Some pesticides have been identified as potential endocrine disrupting chemicals (EDCs); i.e., chemicals that can interfere with the synthesis, secretion, transport, metabolism, binding, and elimination of hormones in the body [5]. The hypothalamus pituitary thyroid axis is one endocrine system that is at risk of experiencing a negative impact due to pesticide exposure [5,6,7]. Hypothyroidism is a condition in which the thyroid gland cannot produce enough hormones (T4 and T3) to meet the body needs [8,9,10]. If a hypothyroid condition occurs in women, it can result in infertility, spontaneous abortion, fetal growth disorders, placental abruption, and prematurely born babies.

Research in Iowa and North Carolina looked at hypothyroidism and hyperthyroidism among female spouses of pesticide applicators (*n* = 16,529) who were enrolled in the U.S. Agricultural Health Study. They assessed the risk of thyroid disease in relation to ever using herbicides, insecticides, fungicides, and fumigants. The prevalence of self-reported clinically diagnosed thyroid disease was 12.5% among those reporting pesticide use, and the prevalence of hypothyroidism and hyperthyroidism was 6.9% and 2.1%, respectively [9]. A study of Indonesian women found an increased risk of thyroid hormone disorders associated with long term pesticide exposure [11].

The Inter-Census Agricultural Survey in 2018 in Indonesia reported that there were 8,051,328 female farmers in Indonesia [12], but little research has been done to investigate the health risks they face as agricultural workers. The objective of this study was to evaluate whether female farmers in one province in Indonesia have significantly different thyroid hormone profiles than female non-farmers in the same region.

## 2. Materials and Methods

### 2.1. Study Population

In Banten Province, where this study was conducted, there were 184,753 female farmers (25% of all farmers) [12]. In 2018, rice productivity in Banten was 1,603,550 tons [13], making this area a regional rice supplier, with 2–3 rice planting periods per year.

This cross-sectional study was assisted by community health workers, health workers from the primary health service, and community leaders in the area. The inclusion criteria included female farmers with children aged newborn to 59 months, without thyroid disease, living in an agricultural area, and participating in the growing of farm crops. We recruited non-agricultural female workers with children aged newborn to 59 months, without thyroid disease from a district in Banten Province which borders the city of Jakarta (this area has few agricultural areas). This study was approved by The Ethical Committee Research Institute YARSI University, (038/KEP-UY/BIA-IV/2019).

### 2.2. Data Collection

The study was conducted from August 2019 to December 2019. Farmers from 3 villages in Pandeglang district were invited to participate in the study and to come to their village hall to receive a health education program from the researchers. With the research team, they filled out questionnaires and underwent physical examinations. The questionnaire consisted of respondent characteristics and risk factors consisting of age, education, occupation, family income, reproductive history (miscarriage, preterm birth, and low birth weight), hand washing habits, vegetable washing habits, working hours, working days, years of work, and use of insecticides at home.

Data on weight, height, BMI and body composition (Tanita model DC-360, Tanita Europe BV, Hoogoorddreef, Amsterdam, The Netherlands), blood pressure (taken twice at 10 min intervals and averaged), spot urine samples, and blood samples were obtained during the physical examination.

Serum was extracted from the blood sample in a non-heparin vacutainer tube stored at −20 °C until analysis. All serum samples were analyzed at YARSI Hospital using an automatic quantitative enzyme immunoassay (Enzyme Linked Fluorescent Assay) on the VIDAS^®^ biomerieux (REF 410417, bioMérieux SA, Marcy-I’Etoile, France) instrument for the determination of free thyroxine (FT4), thyroid stimulating hormone (TSH), triiodothyronine (T3), thyroxine (T4), and free triiodothyronine (FT3). The limit of detection was 0.05 µIU/mL for TSH [14], <0.7 pmol/L for FT3 [15], 0.07 ng/dL for FT4 [16], 0.05 nmol/L for T3 [17], and 37.49 nmol/L for T4 [18].

Urine samples were sent to the Ministry of Health of the Republic of Indonesia Health Laboratory of the Magelang Health Research and Development Agency to measure urinary iodine. This examination used IK-7.2.7 UIE (spectrophotometric method) with a normal range of 100–199 µg/L.

### 2.3. Study Variables

The clinical criteria for high blood pressure used Indonesian guidelines for measuring hypertension (normal values of <90 mmHg for diastole and <140 mmHg for systole) [19]. Body mass index scores were divided into underweight <18.4, normal weight 18.5–25, and overweight >25.1 [20,21]. Measurement criteria for female body fat percentage were based on a normal value of <32% [22]. The normal value of urinary iodine excretion (UIE) was 100–199 µ/L [23]. The measurement of improper hand washing included self-reports of not washing hands in running water using soap. Measurements of insecticide use at home included using mosquito repellent products in the form of lotion, spray, electric plug-in vaporizers, and coils. Agricultural pesticide exposure was defined as “high” for respondents who self-reported that they applied pesticides by spraying, mixed or washed pesticide spray equipment, or applied pesticide tablets or powder.

### 2.4. Statistical Analysis

The descriptive analysis of demographic characteristics and clinical outcomes of thyroid hormone levels was performed using Chi Square and independent t test using SPSS Statistics for Windows, version 18 (SPSS Inc., Chicago, Ill., USA). Comparison of thyroid hormone levels between farmers and non-farmers used a general linear model with the log (base 10) of thyroid hormone levels (TSH, T3, T4, FT3, and FT4) as the outcome, due to the skewed distribution of thyroid levels. The univariate model examined other potential covariates such as age, education level, family income, preterm birth, low birth weight, miscarriage, improper hand washing, number of current work hours, number of current work days, years of work, use of insecticides at home, and dichotomous variables based on clinical normal vs. abnormal levels including body fat percentage, BMI, and blood pressure. When the parameters were significantly associated (*p* value < 0.05) with thyroid levels in univariate models, they were considered for inclusion in the multivariable linear model if they were not colinear. The final models included work hours per day, years of work, and blood pressure.

## 3. Results

### 3.1. Demographics and Risk Factors

The demographics of farmers and non-farmers are shown in Table 1. Most of the respondents were in the age range 26–35 years (61% of farmers and 50% of non-farmers were in this range), and there was no statistically significant difference in age between the groups. The rate of completion of primary school education for farmers was 67%, while 65% of non-farmers received secondary education, resulting in a significant difference in educational level between the groups. Significant differences were also found in family income, where 26% of farmers reported insufficient income and debt, while 48% of non-farmers reported sufficient income and savings. There was a significant difference in the percentage of women reporting premature birth (16% for farmers vs. 6% for non-farmers), but no significant differences in self-reported low birth weight or miscarriages. The working hours per day of farmers and non-farmers were significantly different, with most farmers working fewer than 8 h/day and most non-farmers working more than 8 h a day. Similarly, the working days per week were significantly different between the groups, with almost all non-farmers working ≥5 days per week, while only about half (49%) of farmers worked ≥5 days per week. There was a significant difference between farmers and non-farmers in the years they had been working. Among non-farmers, 28% reported working for <3 years, while only 13% of farmers reported working for <3 years (Table 1).

Farmers were significantly more likely to have abnormal blood pressure than non-farmers, although there was no significant difference in BMI. There was no significant difference in self-reported insecticide use at home, with most of the subjects reporting use (82% of farmers and 74% of non-farmers). However, farmers reported use of insecticides almost daily (68%), whereas 26% non-farmers never used insecticides—a significant difference. Non-farmers were significantly more likely to have an abnormal body fat percentage than farmers. All respondents reported not smoking or drinking alcohol.

Significant differences in the distribution of thyroid level clinical categories were found for FT3 and T4 between farmers and non-farmers. Non-farmers were more likely to be hypo-thyroid for FT3 and hyper-thyroid for T4, while farmers were more likely to be hypo-thyroid for T4 (Table 1). There were no significant differences in the distribution of clinical categories for the hormones FT4, T3, and TSH in farmers and non-farmers. There was a significant difference in the distribution of urinary iodine levels between non-farmers and farmers, with laboratory results showing farmers were more likely to have low urinary iodine levels (Table 1).

Most non-farmers (72%) worked in other people’s homes as household assistants. For farmer respondents, the most common pesticide use frequency was 2–4 times per season for rice, although there were a few female farmers who did not know when pesticides were applied to the rice crops. Almost all farmers carried out planting and harvesting activities, and a few (7–15%) worked directly with pesticides (spraying, applying, and cleaning sprayers) (Table 2).

Most farmers reported that they went to the fields wearing long trousers and long sleeves, although quite a number wore household clothes. Very few reported using any form of personal protective equipment when working in the field.

### 3.2. Comparison of Thyroid Hormones between Farmers and Non-Farmers

A simple comparison between the means of the log (base 10) of thyroid hormone levels for farmers versus non-farmers found significant differences for FT3 and T4, with levels of these hormones being higher in farmers (Table 3).

Multivariable linear regression models for logs (base 10) of FT3, FT4, T3, T4 and TSH were adjusted for other covariates (number of current work hours/day, years of work, and blood pressure). Since these are log linear models, the parameter estimates were exponentiated to become a multiplier, showing that the hormone levels of conventional farmers were significantly higher—i.e., by a factor of 1.06 for FT3 and T4—and significantly lower for farmers for urinary iodine (Table 4).

### 3.3. Pesticides Exposure to Farmers

“High” pesticide exposures were assigned when farmers self-reported performing activities such as pesticide spraying, mixing pesticides, washing spray equipment, or applying pesticide tablets or pesticide powder. In this study, there were 28 farmers with high pesticide exposure. We did not find a significant difference between the clinical distributions of farmer thyroid hormones with high vs. low pesticide exposure (Table 5).

## 4. Discussion

The farmers in this study were more likely to have a low level of education (67%), which was inversely proportional to non-farmers, who were more likely to have a middle level of education (65%). This is in accordance with the research of Kongtip (2018), which showed that conventional and organic farmers in Thailand have a low education level of 57% and 55%, respectively. A quarter of the farmers in this study reported that their income was insufficient for daily life and that they had debt, which was in line with findings in Thailand, where 35% of conventional farmers reported insufficient income with debt [24].

The rate of preterm birth among farmers in this study was quite high (16.5%), and there was a 2.12-fold higher risk of preterm birth among farmers compared to non-farmers. Women farmers in Indonesia still perform activities on the farm until birth. This is consistent with research by Anand (2017), which found that women exposed to pesticides were 1.7 times more likely to give birth to preterm babies than women not exposed to any pesticides [25]. In California, it was reported that exposure to pesticides (e.g., glyphosate, paraquat, imidacloprid) in the first and second trimesters of pregnancy can slightly increase (5–7%) the incidence of preterm birth [26].

We found that the prevalence of hypertension was 33% among farmers and 16% among non-farmers. This is in accordance with the work of Riskesdas (2018), where the prevalence of hypertension in the rural population was 33.7%. This is compared to the population of all farmers in Indonesia, where the prevalence of hypertension as a whole is 36%, while for other workers such as laborers, household assistants, and drivers, the prevalence of hypertension is 30% [27]

The women farmers in this study did not work in the rice fields all day long. This can be seen from the average hours per day, which reached 4.4 h (min–max: 1–9 h per day). They worked from morning to noon in the fields, and after lunch in the fields, they went home to care for their children and the household and did not return to the fields. However, male farmers will continue to work after lunch. When having lunch in the field, 55% of women farmers did not wash their hands properly before eating. From the interviews, many of the farmers washed their hands using water from the fields or used limited quantities of drinking water brought from home and did not use soap. 

In this study, only 15% of female farmers sprayed pesticides, because usually men spray the crops in the fields (Table 4). The women mostly performed planting and harvesting, while some farmers applied fertilizer and removed weeds. Many farmers only wore long sleeves and trousers when working. It was also found that 82% of farmers were exposed to insecticides used in the home. This is because they lived near rice fields and there was a large number of mosquitoes; thus, many of the farmers used lotion (diethyltoluamid (DEET)**,** spray (d-fenotrin, sipermetrin, imiprotrin, praletrin, transflutrin and d-allethrin), plug in vaporizers (d-allethrin, transflutrin), and mosquito coils (piretrum, piretrin, d-allethrin, and esbiotrin) to repel mosquitoes.

Many (36%) of the non-farmers worked ≥8 h per day, while most farmers worked <8 h/day (87%). This was likely due to most of the non-farmers working in other people homes as household assistants, where working hours can be extended. This contrasts with Kongtip’s research (2018), where most agricultural workers in Thailand worked 8–10 h per day [28].

We found that 63% of farmers were hypothyroid for FT3 and 16% were hypothyroid for FT4, but there were few significant differences in the distribution of clinical thyroid measures between the farmers and non-farmers. However, in models that include covariates, FT3 and T4 were significantly higher for farmers compared to non-farmers (Table 4). This can be explained in part by non-farmers being more likely to be hypothyroid for FT3 (Table 1). These findings of higher FT3 and T4 in farmers are in line with the work of Kongtip (2019), which reported an increase in TSH, FT3, T3, and T4 levels in conventional farmers relative to organic farmers in Thailand [6,7]. The higher frequency of hypothyroid for FT3 in non-farmers should be further explored, because a study by Chavez (2013) showed that 11.7% of respondents with complex inflammatory malnutrition syndrome had a low FT3 level, and this was also associated with IL-6, C-reactive protein, and albumin [29]. Low FT3 is included in non-thyroid disease (NTI), which is a counter-regulatory response that causes a waste of energy and protein in acute and chronic stress conditions, such as uncontrolled diabetes mellitus with ketoacidosis, myocardial infarction, and severe myocardial ischemia, and in almost all severe diseases [30,31]. Xue (2017) study found that low FT3 was a worse predictor of health-related quality of life (HRQOL) in Acute Coronary Syndrome (ACS) patients treated with drug-eluting stents (DES) [32]. Blood sampling in this study was conducted in August to December, when the climate was changing to the rainy season. This may have affected their health condition. However, in this study, we did not ask respondents about other health conditions and did not carry out nutritional examinations.

The finding of significant differences in thyroid levels between farmers and non-farming women is important in part because the extent of pesticide exposure does not differ greatly between them. Women farmers in this area generally do not spray pesticides directly. Only 28 (22%) of the farmers reported high pesticide exposure activities. In general, pesticides are not widely used, since only 42% of the farmers reported that pesticides were used in the fields 2–4 times per growing season. In addition, women farmers average only 4.4 h per day and 4.4 working days per week in the fields and primarily perform planting and harvesting. From the interviews conducted, we understand that during busy periods in the fields, such as during planting and harvesting, the women often take their children to the fields while working so that they can perform both childcare and farming responsibilities at the same time.

Perhaps one explanation for the differences between farmers’ and non-farmers’ thyroid levels is the significant difference in the distribution of urinary iodine levels. Laboratory results showed 32% of farmers had low urinary iodine levels and 49% of non-farmers had high iodine levels (Table 1), and even after controlling for covariates, urinary iodine was significantly lower in farmers than in non-farmers (Table 4). Insufficient iodine intake, leading to iodine deficiency, can manifest as hypothyroidism [33,34]. Although our cohort farmers tended to have higher thyroid levels (FT3 and T4) than non-farmers (Table 3 and Table 4), 63% of farmers were considered clinically hypothyroid for FT3 and 16% were clinically hypothyroid for FT4 (Table 1). Selenium, iron, and vitamin A deficiencies can also exacerbate the effects of iodine deficiency since glutathione peroxidase and deiodinase are selenium-dependent enzymes [35]. Another study conducted in Prambanan District, Indonesia, which is an agricultural area and an endemic area for iodine deficiency, found that 15.7% of female respondents aged 18–45 years experienced iodine deficiency [36]. If this condition occurs in pregnant women, it will result in neonatal physical disabilities, mental disabilities, and neurological disorders [23,35].

In this study, we did not ask about food intake; thus, further research is needed to determine the cause of low iodine levels in farmers. Several types of food can interfere with thyroid metabolism (e.g., cabbage, kale, cauliflower, broccoli, cassava, peanuts, sweet potatoes, and others) [35]. In our study, we also found a rate of 16% of preterm birth in farmers. According to Gargari research, iodine deficiency in pregnant women increases the risk of premature birth 3.3-fold [37]. Government iodination programs using iodized salt are encouraged, and iodized salt is inexpensive and easy to obtain in Indonesia.

Meanwhile, for non-farmers, 49% had high urine iodine content. This condition is related to the consumption of foods that are high in iodine. The main sources of iodine include salt, dairy products, and bread, or those naturally abundant in micronutrients, such as seafood [38]. Almost all seafood is high in iodine [35]. In America, the major sources of high-iodine foods come from milk and bread [35,38]. In Japan, seaweed is a popular food. Thyrotoxicosis cases have been widely reported, including one woman who drank tea containing seaweed for 4 weeks [38].

Limitations of this study include the fact that it used only a single cross-sectional measurement of thyroid hormone levels. In addition, we estimated pesticide exposure using a questionnaire for home and agricultural use. We did not include the types of pesticides used, because in the initial survey, almost all the women farmers did not clearly remember the pesticides used. The women only remembered the color and shape of the bottles of pesticides used, so we were not able to verify pesticide use by type. Regarding iodine levels, which can be related to diet, we did not collect food frequency or intake data.

## 5. Conclusions

This study is the first study in Indonesia to examine the association of thyroid hormone levels between women farmers who may be exposed to pesticides and non-farmers. It is of clinical significance that we found that 64% of farmers were hypothyroid for FT3 (16% were hypothyroid for FT4). Despite our hypothesis that the pesticide exposures of farming women would impact thyroid levels when compared to non-farmers, we found few significant differences in the distribution of clinical thyroid measures between farmers and non-farmers. However, simple t-tests and our multivariable models that included covariates showed that FT3 and T4 were significantly higher for farmers compared to non-farmers. Among farmers, 32% had low iodine levels. Although the government has launched an iodination program with iodized salt, there are still many farmers with low urinary iodine, possibly because of the choice of salt that is not iodized or has little iodine, or perhaps this is because the food consumed contains glucosinolates, which interfere with thyroid metabolism, thus exacerbating the effects of iodine deficiency.

To better understand any relationship between pesticide exposure and hormone function, future studies should measure pesticide biomarkers concurrently with target hormone levels.

## Figures and Tables

**Table 1 ijerph-18-06618-t001:** Demographics and risk factors of farmers (*n* = 127) and non-farmers (*n* = 127).

Variables	Farmers	Non-Farmers	*p*-Value	OR	CI
Age					
Mean age (SD)	32.8 (6.620)	31.56 (5.820)			
(Min–max) (year)	(19–47)	(21–44)			
17–25	19 (15.0%)	18 (14.2%)	0.099	NA	
26–35	78 (61.4%)	63 (49.6%)			
36–45	30 (23.6%)	44 (34.6%)			
46–55	0 (0.0%)	2 (1.6%)			
Educational Level					
Low	85 (66.9%)	0 (0.0%)	0.000	NA	
Middle	42 (33.1%)	83 (65.4%)			
High	0 (0.0%)	44 (34.6%)			
Family income					
Enough with savings	25 (19.7%)	61 (48.0%)	0.000	NA	
Enough without savings	69 (54.3%)	65 (51.2%)			
Not enough with debt	33 (26.0%)	1 (0.8%)			
Preterm Birth					
Yes	21 (16.5%)	7 (5.5%)	0.005	2.12	1.105–4.081
No	106 (83.5%)	120 (94.5%)			
Low Birth Weight					
Yes	6 (4.7%)	6 (4.7%)	1	1	0.560–1.785
No	121 (95.3%)	121 (95.3%)			
Miscarriage					
Yes	21 (16.5%)	13 (10.2%)	0.140	0.78	0.579–1.051
No	106 (83.5%)	114 (89.8%)			
Improper hand washing					
Yes	70 (55.1%)	7 (5.5%)	0.000	7.458	3.653–15.223
No	57 (44.9%)	120 (94.5%)			
Current Work hours (hours/day)					
Mean (SD)	4.4 (2.202)	8.27 (6.661)			
(Min-max)	(1–9)	(3–24)			
≥8 h/day	16 (12.9%)	46 (36.2%)	0.000	0.57	0.456–0.709
<8 h/day	111 (87.4%)	81 (63.8%)			
Current Work day (days/week)					
Mean (SD)	4.41 (2.395)	6.24 (0.774)			
(Min-max)	(1–7)	(3–7)			
≥5 days/week	62 (48.8%)	126 (99.2%)	0.000	0.02	0.003–0.159
<5 days/week	65 (51.2%)	1 (0.8%)			
Years of Work					
Mean (SD)	9.47 (6.976)	5.89 (4.172)			
(Min–max)	(1–30)	(1–21)			
≥3 years	110 (86.6%)	92 (72.4%)	0.005	1.48	1.160–1.883
<3 years	17 (13.4%)	35 (27.6%)			
BMI (kg/m^2^)					
Underweight	6 (4.7%)	9 (7.1%)			
Normal	65 (51.2%)	63 (49.6%)	0.726		
Overweight	56 (44.1%)	55 (43.3%)			
Blood Pressure (mmHg)					
Abnormal (≥140 and ≥90)	42 (33.1%)	20 (15.7%)	0.001	1.73	1.179–2.532
Normal (<90 and <140)	85 (66.9%)	107 (84.3%)			
Systolic (mmHg)					
Abnormal (<90 and ≥140)	12 (9.4%)	5 (3.9%)	0.079	1.75	0.829–3.693
Normal (90–<140)	115 (90.6%)	122 (96.1%)			
Diastolic (mmHg)					
Abnormal (<65 and ≥90)	85 (66.9%)	21 (16.5%)	0.000	3.61	2.433–5.373
Normal (65–<90)	42 (33.1%)	106 (83.5%)			
Body Fat %					
Abnormal ≥ 32%	78 (61.4%)	96 (75.6%)	0.015	0.732	0.576–0.930
Normal < 32%	49 (38.6%)	31 (24.4%)			
Insecticide used in home					
Yes	104(81.9%)	94 (74.0%)	0.130	1.24	0.954–1.615
No	23 (18.1%)	33 (26.0%)			
Frequency of insecticide use in home					
>5 day/a week	86 (67.7%)	35 (27.6%)			
2–4 day/a week	10 (7.9%)	8 (6.3%)			
1 day/a week	2 (1.6%)	13 (10.2%)	0.000	NA	
1–3 times/month	2 (1.6%)	13 (10.2%)			
<1 time/month	4 (3.1%)	25 (19.7%)			
Never	23 (18.1%)	33 (26.0%)			
Hormone FT3					
Hypo < 4	81 (63.8%)	116 (91.3%)	0.000 *	NA	
Normal (4–8.3)	46 (36.2%)	9 (7.1%)			
Hyper > 8.3	0 (0.0%)	2 (1.6%)			
Hormone FT4					
Hypo < 10.6	20 (15.7%)	15 (11.8%)	0.646 *	NA	
Normal (10.6–19.4)	106 (83.5%)	109 (85.8%)			
Hyper > 19.4	1 (0.8%)	3 (2.4%)			
Hormone T3					
Hypo < 0.92	6 (4.7%)	5 (3.9%)	0.446 *	NA	
Normal (0.92–2.33)	119 (93.7%)	116 (91.3%)			
Hyper > 2.33	2 (1.6%)	6 (4.7%)			
Hormone T4					
Hypo < 20	1 (0.8%)	5 (3.9%)	0.018 *	NA	
Normal (60–120)	125 (98.4%)	119 (93.7%)			
Hyper > 120	1 (0.8%)	3 (2.4%)			
Hormone TSH					
Hypo < 0.25	0 (0.0%)	2 (1.6%)	0.568 *	NA	
Normal (0.25–5)	126 (99.2%)	125 (98.4%)			
Hyper > 5	1 (0.8%)	0 (0.0%)			
Urine Iodine					
Hypo < 100 µ/L	41(32.3%)	15 (11.8%)	0.000	NA	
Normal (100–199 µ/L)	46 (36.2%)	50 (39.4%)			
Hyper > 199 µ/L	40(31.5%)	62(48.8%)			

* Independent *t* test.

**Table 2 ijerph-18-06618-t002:** Kinds of work for farmers (*n* = 127).

Work	Yes
Planting	103 (81.1%)
Fertilization	36 (28.3%)
Apply pesticide tablets	9 (7.1%)
Application of pesticide powder	18 (14.2%)
Spraying pesticides	19 (15.0%)
Mixing pesticides	15 (11.8%)
Wash the sprayer	19 (15.0%)
Remove the weeds	65 (51.2%)
Harvest	102 (80.3%)

**Table 3 ijerph-18-06618-t003:** Thyroid hormones for farmers (*n* = 127) and non-farmers (*n* = 127).

Thyroid Hormones	Farmers	Non-Farmers	Mean * Difference (95% CI of the Difference)	*p*-Value *
FT3 (µIU/mL)				
Geometric mean	1.61	1.52	0.94 (0.92–0.96)	0.000
Min–max	1.35–1.90	1.24–2.65		
FT4 (µIU/mL)				
Geometric mean	2.52	2.56	1.01(0.10–1.03)	0.118
Min–max	2.14–2.99	2.01–3.44		
T3 (µIU/mL)				
Geometric mean	1.15	1.14	0.99 (0.97–1.02)	0.528
Min–max	0.90–1.37	0.93–1.75		
T4 (µIU/mL)				
Geometric mean	5.02	4.63	0.92 (0.87–0.97)	0.003
Min–max	4.41–5.76	0.95–6.37		
TSH (µIU/mL)				
Geometric mean	1.14	1.01	0.96 (0.91–1)	0.104
Min–max	0.62–2.10	0.43–1.75		

* Unpaired *t* test of the log (base 10) concentrations.

**Table 4 ijerph-18-06618-t004:** Comparison of log (10) thyroid hormone levels for farmers versus non-farmers using a generalized linear model *.

Thyroid Hormones	Expβ (95% CI)
FT3	1.056 (1.030–1.084)
FT4	0.984 (0.966–1.004)
T3	1.016 (0.989–1.043)
T4	1.058 (1.027–1.091)
TSH	1.023 (0.974–1.073)
Iodine Urine	0.866 (0.811–0.925)

* All models controlled for work hours per day, years of work, and blood pressure.

**Table 5 ijerph-18-06618-t005:** Pesticides exposure vs. hormones and urine iodine in farmers (*n* = 127).

Thyroid Hormones	Pesticides Exposure		*p*-Value
	High (*n* = 28)	Low (*n* = 99)	
Hormone FT3			
Hypo < 4	18 (64.3%)	63 (63.6%)	0.950
Normal (4–8.3)	10 (35.7%)	36 (36.4%)	
Hormone FT4			
Hypo < 10.6	3 (10.7%)	17 (17.2%)	0.343 *
Normal (10.6–19.4)	25 (89.3%)	81 (81.8%)	
Hyper >19.4	0 (0.0%)	1 (1.0%)	
Hormone T3			
Hypo < 0.92	2 (7.1%)	4 (4.0%)	0.857 *
Normal (0.92–2.33)	26 (92.9%)	93 (93.9%)	
Hyper > 2.33	0 (0.0%)	2 (2.0%)	
Hormone T4			
Normal (60–120)	28 (100%)	98 (99.0%)	1 **
Hyper > 120	0 (0.0%)	1 (1.0%)	
Hormone TSH			
Normal (0.25–5)	28 (100%)	98 (99.0%)	1 **
Hyper > 5	0 (0.0%)	1 (1.0%)	
Urine Iodine			
Hypo < 100	9 (32.1%)	32 (32.3%)	0.257
Normal (100–199)	7 (25.0%)	39 (39.4%)	
Hyper > 199	12 (42.9%)	28 (28.3%)	

* Mann Whitney for *p*-value, ** Fisher test.

## Data Availability

Data from this study are available from the corresponding author on reasonable request.

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
