# Peer review of "Comparison of Thyroid Hormone Levels between Women Farmers and Non-Farmers in Banten Indonesia"

_ijerph, 2021, doi:10.3390/ijerph18126618_

Round 1

Reviewer 1 Report

The study is of interest, however, the title does not reflect what has been done.
It is about comparison of thyroid hormone and iodine levels between two groups of women: farmers and non-farmers.
No pesticide measurement or exposure is made, it is assumed that those who work in agriculture are exposed.

I suggest to change the title:
Comparison of Thyroid Hormone Levels between Women Farmers and Non-farmers in Banten Indonesia.

The authors do not mention the design of the study...
It looks to me that is a transversal study comparing two groups, asuming exposed and non-exposed, according to the occupation.
There is no measurement of exposure. This may be a limitation, because there are some settings in where non-farmers, but living in areas around farming, are also exposed to pesticides.
I suggest so check: https://doi.org/10.1371/journal.pone.0196084

Table 2: there is no need to include the NO column.

Discussion:
Lines 249-253: Maybe, there is a gender issue? The info of Thailand as a whole may be included both gender?
Line 262: by Chavez is in a bigger font.
Line 272-277: Would be inportant to share in which season was done the study, because it could affect th results.

Conclusions:
A reflection: Is it possible that the differences in thyroid factors and iodine are due to living and dietary conditions,
rather than exposure to pesticides?
Format: Bibliographic citations must go together in parentheses when there is more than one. The format of citations and references does not correspond to the one used by the journal

Author Response

Dear Reviewer,

Thank you for the advice. This is the revision.

I have changed the title of the research and added the research type in the second paragraph of 2.1. Study population.

There was no measurement of pesticide exposure among non-farmer respondents because they live far from agricultural areas even though they live in one province. Hopefully they didn't come in contact with pesticides in agriculture.

Column "NO" in table 2 has been removed.

I added the phrase "Blood sampling in this study was conducted in August-December, when the climate was changing to the rainy season. This may affect their health condition." at the result.

I've changed the bibliography citation.

Warm regards,

Dian Mardhiyah

Reviewer 2 Report

The work is very interesting but the active ingredients of the pesticides and the quantities used to make a correct comparison are missing.
In the conclusions it should be explained that in order to have more interesting results, research in this sense should be done  

Author Response

Dear Reviewer,

Thank you for the advice.

I have added the sentence below at the conclusion

"However there is a need for further research measuring the presence of the active ingredient and quantities used for pesticide exposure among female farmers."

Warm regards,

Dian Mardhiyah

Reviewer 3 Report

    1. Adding experimental photos to the text can make the data source more convincing, and visualize the analyzed data, which helps readers understand the content of the text.

    2. It was written at the beginning of the article that some pesticides have been identified as potential  endocrine disrupting chemicals, so pesticides will affect the level of thyroid hormones in female farmers. If more analysis of pesticide types is added, it can be more research significant.

    3. The paper discusses the problem of iodized salt intake, but the lack of actual investigations on iodized salt intake is not logically rigorous.

Author Response

Dear Reviewer,

Thank you for the advice.

  1. There are no photos that I can add
  2. At the beginning of the research I added a question for the type of pesticide used, but almost all women farmers did not remember either the brand or the active substance of the pesticide used in the workplace. They only remember the color and shape.
  3. "In this study we did not ask about food intake, further research is needed to determine the cause of low iodine levels in farmers. Several types of food can interfere with thyroid metabolism (eg cabbage, kale, cauliflower, broccoli, cassava, peanuts, sweet potatoes and others)." (revised sentence in the tenth paragraph at the "conclusion")

Warm regards,

Dian Mardhiyah

Round 2

Reviewer 1 Report

The authors made the suggested changes 

Author Response

Thank you very much